# Nosocomial Transmission of Emerging Viruses via Aerosol-Generating Medical Procedures

**DOI:** 10.3390/v11100940

**Published:** 2019-10-12

**Authors:** Seth D. Judson, Vincent J. Munster

**Affiliations:** 1Department of Medicine, University of Washington, Seattle, WA 98195, USA; sethdjudson@gmail.com; 2Laboratory of Virology, Division of Intramural Research, National Institute of Allergy and Infectious Diseases, National Institutes of Health, Hamilton, MT 59840, USA

**Keywords:** aerosols, viruses, cross infection, infection control, risk, health personnel, virus diseases, hemorrhagic fevers, viral, ebolavirus

## Abstract

Recent nosocomial transmission events of emerging and re-emerging viruses, including Ebola virus, Middle East respiratory syndrome coronavirus, Nipah virus, and Crimean–Congo hemorrhagic fever orthonairovirus, have highlighted the risk of nosocomial transmission of emerging viruses in health-care settings. In particular, concerns and precautions have increased regarding the use of aerosol-generating medical procedures when treating patients with such viral infections. In spite of increasing associations between aerosol-generating medical procedures and the nosocomial transmission of viruses, we still have a poor understanding of the risks of specific procedures and viruses. In order to identify which aerosol-generating medical procedures and emerging viruses pose a high risk to health-care workers, we explore the mechanisms of aerosol-generating medical procedures, as well as the transmission pathways and characteristics of highly pathogenic viruses associated with nosocomial transmission. We then propose how research, both in clinical and experimental settings, could advance current infection control guidelines.

## 1. Introduction

Aerosol-generating medical procedures (AGMPs) are increasingly being recognized as important sources for nosocomial transmission of emerging viruses. Intubation was investigated as a possible cause of Ebola virus (EBOV) transmission among health-care workers (HCWs) in the United States [1]. Additionally, the high rate of nosocomial transmission of Middle East respiratory syndrome coronavirus (MERS-CoV) and Severe Acute Respiratory Syndrome coronavirus (SARS-CoV) caused speculation about the role of AGMPs [2,3,4,5]. Crimean–Congo hemorrhagic fever orthonairovirus (CCHFV), was also associated with nosocomial infection secondary to AGMPs [6]. While guidelines were developed for performing AGMPs on patients with certain viral infections, assessing and understanding the risk that specific viruses and AGMPs pose for nosocomial transmission could improve infection control practices, as well as reveal relationships in virus transmission.

Despite the perceived importance of AGMPs in nosocomial transmission of viruses and other infectious agents, scarce empirical or quantitative evidence exists [7]. In order to assess the risk that certain viruses and AGMPs create for nosocomial transmission, we first need to identify potential AGMPs and viruses. The second step is then to determine the risk associated with these viruses and procedures, either through retrospective analysis, investigating the circumstances of nosocomial transmission, or through experiments, such as using air sampling during AGMPs to determine the risk of generating infectious virus-laden aerosols. Lastly, we can use this knowledge to re-evaluate current guidelines and communicate which viruses and AGMPs pose the highest risk for nosocomial transmission.

## 2. Aerosol-Generating Medical Procedures

Medical procedures that have the potential to create aerosols in addition to those that patients regularly form from breathing, coughing, sneezing, or talking are called AGMPs [7]. While there are many suspected AGMPs, few AGMPs were confirmed to generate aerosols. In order to determine which AGMPs could be important for nosocomial virus transmission, we first need to characterize what aerosols are and how they are created. 

Aerosols are particles suspended in air that can contain a variety of pathogens, including viruses, and there is ongoing debate about how to classify them. Many divide aerosols into the categories of small droplets (which some exclusively call aerosols) and large droplets, with small droplets having the potential to desiccate and form droplet nuclei that travel long distances, while large droplets do not evaporate before settling on surfaces [8]. Classifying aerosols by their initial size is relevant in relation to their dispersal patterns, but it is also important to classify aerosols according to where they deposit in the respiratory tract because pathogenesis can be influenced by whether a virus deposits in the upper respiratory tract (URT) or lower respiratory tract (LRT) [9]. Dispersal and deposition depend on a variety of factors, and there is no exact cutoff for small and large droplets. Some authors use ≤5 µm in diameter as a cutoff for small droplets, while another possible cutoff between aerosol types is 20 µm, since aerosols ≤20 µm in diameter can desiccate to form droplet nuclei, and aerosols ≥20 µm do not deposit substantially in the LRT [10].

Often the term airborne transmission is used to describe infection by small droplet aerosols and droplet nuclei, while droplet transmission refers to the route of large droplet aerosols. Since aerosols can be of multiple sizes, we use the term aerosol transmission to generally describe transmission through the generation of infectious small and large droplet aerosols. In addition to these modes of transmission, AGMPs may also create opportunities for direct contact and fomite transmission, which may be difficult to distinguish.

HCWs are considered to be at risk for nosocomial virus transmission from both small and large droplet aerosols, for both seem to play a role in human-to-human virus transmission. Small droplets can be inhaled into the LRT, while large droplets can splash into the eyes or mouth and deposit in the URT. Certain respiratory viruses, like influenza A virus, are believed to transmit between people by both small and large droplets [10], whereas other nonrespiratory viruses, like EBOV, could theoretically be spread by large droplets because small droplets containing these viruses are not known to form in the human respiratory tract [8]. It is unknown whether certain AGMPs generate either small or large droplets, or both. Therefore, depending on what aerosols are formed, AGMPs could potentially amplify a normal route of transmission for respiratory viruses or open up a new route of transmission for other viruses. 

We can group possible AGMPs into two categories: procedures that mechanically create and disperse aerosols and procedures that induce the patient to produce aerosols (Figure 1 and Table 1). Procedures that irritate the airway, such as bronchoscopy or tracheal intubation, can cause a patient to cough forcefully, potentially emitting virus-laden aerosols, and both of these procedures are associated with the possibility of increasing the risk of SARS-CoV transmission among HCWs [11,12]. The pressure on a patient’s chest during cardiopulmonary resuscitation can also induce a “cough-like force”, which was another possible source of SARS-CoV nosocomial transmission [13]. Sputum is also routinely collected from patients for diagnostic purposes by cough induction, but it is not associated with nosocomial virus transmission.

In contrast to causing a patient to produce aerosols, AGMPs can also mechanically create and disperse respiratory aerosols through procedures such as ventilation, suctioning of the airway, or nebulizer treatment. Both manual ventilation, using a bag-valve-mask, and other forms of noninvasive ventilation (NIV), such as continuous positive airway pressure (CPAP), bilevel positive airway pressure (BiPAP), and high-frequency oscillatory ventilation (HFOV) are associated with SARS-CoV nosocomial transmission [11]. Although the exact mechanisms of how these procedures create virus-laden aerosols in the respiratory tract remain unknown, it is possible that forcing or removing air from the respiratory tract could generate aerosols. 

While AGMPs are traditionally thought of in regard to the generation of respiratory aerosols, AGMPs can also aerosolize infected fluids in other regions of the human body. Surgical techniques can aerosolize blood and possibly viruses. For example, infectious HIV-1 was found in the aerosols generated by surgical power tools [14], and a tracheotomy was associated with SARS-CoV transmission [11]. Lasers can create plumes of debris that contain infectious aerosolized virus, as well [15]. It is important to recognize the range of AGMPs and the circumstances under which they might be performed on infected patients. In order to associate certain AGMPs with nosocomial virus transmission, researchers need to test whether certain procedures generate aerosols with infectious virus, either through hospital sampling or laboratory procedures.

## 3. Emerging High-Risk Viruses

Knowing the mechanisms behind different AGMPs allows us to narrow our focus of emerging viruses that could cause nosocomial transmission via AGMPs. These viruses must be able to opportunistically infect via the aerosol route and must be present in the patient where the AGMP is taking place. These two conditions fit a wide spectrum of viruses, and therefore we focus on those that are emerging and pose a high risk to HCWs performing AGMPs. Here we define high risk as both a high likelihood of infection if an aerosol is inhaled or comes into contact with a mucous membrane and a high case-fatality rate for the viral disease. Such viruses are those that are highly infectious and pathogenic and for which limited prophylactic or therapeutic countermeasures are available. This includes most biosafety level 3 and 4 viruses, while it excludes viruses such as the measles, mumps, and rubella, which can infect via the aerosol route and be spread by AGMPs, but a common vaccine protects HCWs against them. Viruses such as Norwalk virus, enteroviruses, or human respiratory syncytial virus (RSV), which either cause self-limiting diseases or are primarily pathogenic in pediatric, pregnant, or immunocompromised patients, are also included. Infection control measures must be performed on these and other viruses, for other patients and hospital visitors are also at risk of nosocomial virus transmission [16]. However, here we focus on novel, high-risk viruses for HCWs performing AGMPs.

When considering emerging highly pathogenic viruses, we used the following criteria to assess which viruses are high risk for HCWs performing AGMPs. (1) The virus is infectious via small or large droplet aerosols in humans or non-human primates (NHPs). Although some virus-laden aerosols, specifically large droplets, could land on surfaces and cause subsequent fomite transmission, those that can infect via inhalation of aerosols or aerosol contact with the eyes or mucous membranes may have the most potential to cause nosocomial transmission through AGMPs. (2) The virus is found in the human respiratory tract or in the respiratory tract of NHPs. Most AGMPs occur in the upper or lower human respiratory tract, and, therefore, if the virus is present in these locations, we expect that it has a higher likelihood of becoming aerosolized. (3) There is previous evidence of nosocomial transmission or association with AGMPs. This third criteria identifies viruses that were associated with this type of transmission in the past; although, it may miss viruses that are emerging or do not frequently infect humans.

The viruses that pose the highest risk to HCWs performing AGMPs may be some of the viruses that we know the least about. This is because many of these viruses may be emerging zoonotic viruses that rarely infect humans compared to human-adapted viruses. Therefore, we relied on knowledge from both human and animal viral infections to create a list of potential viruses (Table 2). We included viruses that fit at least two of the three previously mentioned classifications. Many of these viruses are also on the World Health Organization’s list of priority pathogens for research and development preparedness [17].

The viruses that we identify as high risk come from eight families and have diverse characteristics. However, they also have much in common. All of these viruses are emerging or re-emerging zoonotic RNA viruses. Initially spilling over from animal hosts into humans, these viruses can undergo subsequent human-to-human and nosocomial transmission to cause epidemics. However, unlike viruses that have evolved with human hosts, none of these viruses are endemic in human populations. Some of these viruses may cause stuttering chains of transmission before disappearing from human populations, while others, like Ebola virus or pandemic influenza A virus, may cause large outbreaks before being contained or establish themselves as endemic pathogens in the human population [18]. Viruses evolve within their hosts to maintain an optimum balance between transmission (consequently virulence) and persistence [19]. Because humans are not the reservoir hosts for these viruses, this equilibrium was not established. Most of these high-risk viruses are highly infectious and virulent, but are not as efficient at transmitting and persisting as true human respiratory viruses. Of the high-risk viruses, those belonging to the families of coronaviruses, orthomyxoviruses, and paramyxoviruses come closest to the equilibrium of persistence and transmissibility in humans. Uncoincidentally, these viral families also contain viruses that have become endemic in human populations.

Considering these eight viral families, we can begin to assess the risk of particular viruses. To make a conclusive risk assessment, we need further experimental and epidemiological evidence. We can inform these future studies by first looking at what is currently known about the viruses in each of these families in regard to AGMPs.

### 3.1. Bunyavirales: Arenaviridae

The family Arenaviridae, in the order Bunyavirales, contains multiple viruses that have the potential for nosocomial transmission due to AGMPs. Arenaviruses spill over when humans inhale the aerosolized excreta of the rodent hosts for these viruses. Additionally, laboratory workers and NHPs were also infected by aerosolized arenaviruses, specifically Junin virus, Lassa virus, and Machupo virus [23]. Arenavirus infections can cause viral hemorrhagic fevers (VHFs) in humans. Known nosocomial transmission has occurred from patients infected with Lassa virus and Machupo virus [26,27]. Less clinical or experimental data exist for other arenaviruses, but these may have similar characteristics. Although arenaviruses are found in the respiratory tracts of humans or animal models [24,25,28], there is no evidence for human-to-human airborne transmission of arenaviruses. However, arenavirus aerosol transmission could theoretically occur when aerosols are mechanically generated through AGMPs.

### 3.2. Bunyavirales: Hantaviridae, Nairoviridae, Phenuiviridae

Like arenaviruses, multiple other viruses within the order Bunyavirales are zoonotic and can cause VHFs. Both Crimean–Congo hemorrhagic fever (CCHF) orthonairovirus and Andes hantavirus were associated with nosocomial transmission, with CCHF transmission occurring after AGMPs were performed without eye or respiratory protection [6,31]. Interestingly, other hantaviruses were aerosol transmitted to humans, either through accidentally aerosolizing the virus or through inhaling aerosolized rodent excreta [23], but only Andes hantavirus was confirmed to transmit from person to person [31]. Although hantaviruses can cause respiratory disease in humans, as in hantavirus pulmonary syndrome (HPS), and were found in the respiratory tract [30], they do not seem to efficiently transmit between humans like typical respiratory viruses. However, similar to arenaviruses, certain bunyaviruses found in human- or animal-model respiratory tracts, including CCHF virus, hantaviruses, and Rift valley fever virus, could possibly gain the route of aerosol transmission if an AGMP is performed.

### 3.3. Coronaviridae

The family of Coronaviridae contains viruses that are known to transmit routinely between humans through the aerosol route. Both MERS-CoV and SARS-CoV cause respiratory disease in humans and transmit via aerosols, but it is unknown whether small-droplet or large-droplet aerosols are the modes of transmission for these viruses. Significant nosocomial transmission of SARS-CoV has incited the most research regarding the role of AGMPs in nosocomial virus transmission [11,12,13], while recent nosocomial transmission events of MERS-CoV warrant further research [2,3,4,5]. Therefore, AGMPs could possibly amplify an already established route of infection for these viruses.

### 3.4. Filoviridae

Filoviruses also cause VHFs and could potentially become transmissible through AGMPs. During outbreaks of Ebola virus disease (EVD), concerns were raised regarding airborne transmission of EBOV because of nosocomial transmission events and the discovery of the virus in the human respiratory tract [36,38]. Given what we know from experimental and epidemiological evidence, airborne transmission is unlikely, yet it is possible that AGMPs could create infectious EBOV-laden aerosols that could lead to nosocomial transmission [8,44]. Likewise, viruses belonging to the other species of ebolaviruses that are pathogenic in humans, as well as another filovirus, Marburg virus, could share similar transmission properties.

### 3.5. Orthomyxoviridae

The family of viruses Orthomyxoviridae contains both human and zoonotic viruses, of which the most well-known are influenza viruses. While seasonal flu vaccines protect HCWs against influenza viruses currently circulating in the human population (influenza A virus H1N1, influenza A virus H3N2, and influenza B virus), HCWs are not safe from novel influenza A virus subtypes emerging from animals, namely avian and swine influenza A viruses. This includes influenza A virus subtypes such as H5N1 and H7N9. Some influenza A viruses are known to transmit between humans via aerosols [10]. However, there is variation in the efficiency of human-to-human aerosol transmission of different influenza A viruses. For instance, to date, there were few instances of nosocomial transmission of H7N9 or H5N1, even when protective measures were not used, while there were multiple nosocomial transmission events of pandemic H1N1 and other influenza A virus subtypes [39,45,46,47,48,49]. This could be due to variation in infectivity of the virus and tissue tropism, with the pandemic H1N1 preferentially replicating in the human upper respiratory tract and the avian influenza A viruses preferentially replicating in the lower respiratory tract [50,51]. Therefore, AGMPs have the potential to amplify or open up the route of aerosol transmission for influenza A viruses. Understanding the characteristics of the aerosol transmission of different influenza A viruses and subtypes could help us determine the risk of influenza A viruses for nosocomial transmission due to AGMPs. 

### 3.6. Paramyxoviridae

Of the viruses in the family Paramyxoviridae, the recently emerged Nipah and Hendra viruses pose a high risk to HCWs performing AGMPs. Both viruses are known to cause respiratory disease in humans, and nosocomial transmission was documented for Nipah virus [42]. Aerosol transmission is one of the suspected routes of transmission for Nipah virus because of evidence from contact tracing and finding the virus in human respiratory secretions [41]. Hendra virus spilled over from horses to humans, and no human-to-human transmission was documented, but the virus was detected in human lungs [43]. Although henipaviruses do not seem to be as contagious through the aerosol route between people as other viruses in their family, such as measles virus, it is possible that AGMPs could contribute to the formation of infectious Nipah- and Hendra-laden aerosols that could cause nosocomial infection.

## 4. Designing Experiments and Assessing Risk

Given the lack of experimental or epidemiological data on nosocomial virus transmission and AGMPs, current guidelines for infection control are based on the precautionary principle. In order to have a more nuanced understanding of the risks associated with difference AGMPs and viruses, we need more research in both clinical and experimental settings. This type of research could take two different forms, retrospective epidemiological studies or on-site sampling and experimental tests. The former was used to assess the risk of AGMPs and SARS-CoV transmission [11]. However, the quality of retrospective data limits these kinds of studies. Control cases are necessary to rule out other sources of nosocomial transmission besides AGMPs, such as direct patient contact and fomite transmission. Moreover, these studies rely on reporting that may be infrequent and/or unreliable.

Air sampling for viruses during AGMPs performed on patients would provide the most clinically relevant data. Multiple air-sampling techniques for viruses now exist. Researchers can use both solid and liquid impactors to sample and recover aerosolized viruses [52]. Personally worn bioaerosal samplers and stationary room samplers were used to detect influenza A virus RNA in an emergency department [53]. Researchers also used aerosol samplers to determine the amount of influenza A (H1N1) RNA in aerosols in the vicinity of patients while AGMPs were being performed [54]. This allowed the authors to determine which procedures were associated with a higher concentration of viral RNA [54]. In order to determine whether viable virus is present in aerosols, virus isolation could be performed from air samples, and further quantification could occur through titrations or plaque assays [52]. One group used a simulated aerosol chamber to demonstrate that viable influenza virus A could be extracted from surgical masks and N95 respirators [55]. Particle sizers may also be used in experimental settings to characterize the size and dispersal of aerosols. One group used particle sizers to measure the size and travel distance of aerosols from patients who underwent AGMPs, such as nebulizer treatment and NIV [56]. Determining the quantity of viable virus expelled from certain patients during AGMPs could help determine phenomena like super-spreading events, while understanding aerosol characteristics such as particle size could elucidate mechanisms of transmission.

While on-site sampling works for current nosocomial transmission events, we can design experiments to gain prospective knowledge. Procedures such as bronchoscopy and intubation are performed on animal models of the high-risk viral diseases we identified, and air sampling during these procedures could determine whether they are aerosol-generating. Experimentally generating virus-laden aerosols of different sizes and under different environmental conditions could also help determine the risks of different viruses based on their stability in aerosols [57,58]. Researchers could then create risk models for different viruses based on aersol stability, as well as data on the quantity, concentration, travel distance, and size of aerosols formed during AGMPs.

Multiple environmental factors influence the viability of aerosolized viruses, including relative humidity, temperature, UV radiation, and gas composition of the air [52]. These factors affect viruses differently, and so it is important to consider the environments where emerging viruses exist and where AGMPs are being performed. The aforementioned AGMPs include those that are both ubiquitous, such as CPR or manual ventilation, as well as those that are limited to advanced health-care settings, such as bronchoscopy. The risk of transmission from AGMPs may be very different in a field-based treatment unit than from a tertiary care hospital. Additionally, some health-care settings, such as hospitals with biocontainment units, include additional air-handling systems that can limit aerosol exposure to HCWs [59]. Therefore, the risk of nosocomial transmission via AGMPs varies greatly based on the environment, and this must be considered when designing experiments.

## 5. Discussion

During the 2013–2016 outbreak of EVD, the Centers for Disease Control and Prevention (CDC) updated its guidelines regarding precautions to prevent the transmission of EBOV in health-care settings [60]. The updated guidelines further emphasized proper personal protective equipment (PPE) and isolation when performing AGMPs on EVD patients. Currently, in addition to standard PPE, the CDC recommends the use of eye protection, airborne infection isolation rooms, and N95 or higher respirators when performing AGMPs on patients with VHFs, SARS-CoV, and avian or pandemic influenza A viruses [61]. The CDC made similar recommendations as an interim guidance for MERS-CoV [62], and we were unable to find additional CDC AGMP guidelines for Nipah virus, Hendra virus, or hantaviruses. More evidence of nosocomial transmission events of these and other viruses due to AGMPs is likely to prompt future guidelines. Therefore, our understanding of the risks associated with AGMPs and nosocomial virus transmission is not static, and we must continue to improve our knowledge to develop appropriate precautions.

The ambiguity of which procedures and viruses require additional protective measures during AGMPs may lead to breaches in protocol. During many of the cited nosocomial transmission events, HCWs did not use proper eye or respiratory protection. Even when aware of the need for respiratory protection, HCWs may mistakenly wear surgical masks or unfitted N95 respirators, which do not provide proper protection. Additionally, HCWs may not have access to approprtiate PPE depending on the health-care environment. Therefore, while determining the risks of certain viruses and procedures is essential, communicating their respective precautions and providing resources is equally important. Likewise, proper patient triage and diagnosis are the first steps to ensuring that precautions are undertaken when performing AGMPs. 

Overall, more research and communication about the risks of certain viruses and AGMPs are necessary to resolve the uncertainty surrounding their role in nosocomial virus transmission. Although we identified certain viruses and procedures that could be high risk and should be experimentally or clinically tested, emerging viruses or novel procedures may also play significant roles. If we are to design proactive infection control guidelines and understand the underlying biology of viral transmission, we must conduct collaborative clinical and scientific research on nosocomial virus transmission and AGMPs.

## Figures and Tables

**Figure 1 viruses-11-00940-f001:**
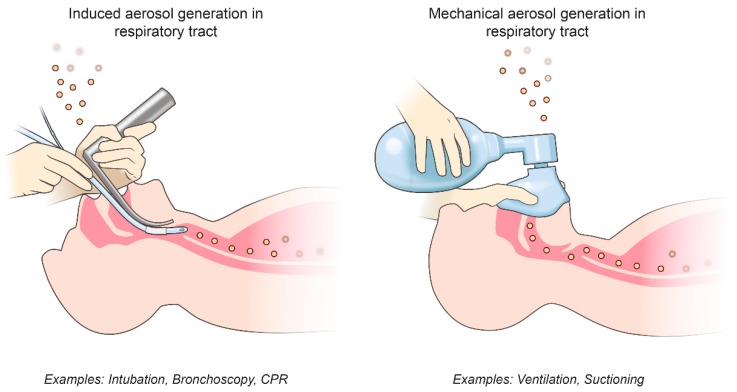
Potential types of aerosol-generating medical procedures (AGMPs). AGMPs can be divided into procedures that induce the patient to produce aerosols and procedures that mechanically generate aerosols themselves.

**Table 1 viruses-11-00940-t001:** Potential aerosol-generating medical procedures involved in nosocomial virus transmission.

AGMP	How/Where Aerosols May Be Generated
Bronchoscopy *	Induced cough, respiratory tract
Cardiopulmonary resuscitation *	Induced cough, respiratory tract
Noninvasive ventilation * (BiPAP, CPAP, HFOV)	Possible mechanical dispersal of aerosols, respiratory tract
Tracheal intubation *	Induced cough, respiratory tract
Manual ventilation *	Possible mechanical dispersal of aerosols, respiratory tract
Surgery	Cutting bone and tendon, and irrigation aerosolize blood
Sputum induction	Induced cough, respiratory tract
Nebulizer treatment	Possible mechanical dispersal of aerosols, respiratory tract
Suctioning	Possible mechanical dispersal of aerosols, respiratory tract
Laser plume	Mechanical dispersal of aerosols

* Possible association with SARs-CoV transmission [11,12,13].

**Table 2 viruses-11-00940-t002:** Emerging viruses that may pose a high risk to health-care personnel when performing aerosol-generating medical procedures.

Family, Virus	Infectious via Aerosol?	Evidence in Respiratory Tract?	Nosocomial Transmission/AGMPs?
*Arenaviridae* i. Junin virus ii. Lassa virus iii. Machupo virus	i, ii, iii. NHPs infected by aerosol administration [20,21,22] ii. Laboratory workers infected by inhaling aerosols [23]	Upper respiratory: i. Viral RNA in NHP oral and nasal swabs [24] ii. Virus isolated from human throat swabs [25]	ii, iii. Nosocomial transmission [26,27]
		Lower respiratory: ii. Virus isolated from human lung [28]	
*Hantaviridae, Nairoviridae, Phenuiviridae* i. CCHF virus ii. Hantaviruses iii. Rift valley fever virus	i, ii, iii. Laboratory workers infected by inhaling aerosols [23,29]	Upper respiratory: i, iii. Viral RNA in NHP nasal swabs [24] iii. Viral RNA in NHP oral swabs [24] ii. Viral RNA in human saliva [30] Lower respiratory:	i, ii. Nosocomial transmission [6,31] i. Likely association with AGMPs [6]
		ii. Causes human respiratory disease. Viral antigen in human lung [32]	
*Coronaviridae* i. MERS-CoV ii. SARS-CoV	ii. Laboratory and health-care workers infected by inhaling aerosols [23] i, ii. Known human-to-human aerosol transmission	Upper respiratory: ii. Viral RNA in nasal/throat swabs [33]	i, ii. Nosocomial transmission [2,11,12] ii. Significant association with AGMPs [11,12,13]
		Lower respiratory: i, ii. Causes human respiratory disease. Virus isolated from lung and sputum [33,34]	
*Filoviridae* i. Ebolaviruses ii. Marburg virus	i, ii. NHPs infected by aerosol administration [35]	Upper respiratory: i. Isolated from human saliva [36] ii. Isolated from NHP saliva [37]	i. Nosocomial transmission [1] i. Possible association with AGMPs [1]
		Lower respiratory: i. Virus particles detected in human alveolar space [38]	
*Orthomyxoviridae* i. Influenza A virus (H5N1, H7N9, pandemic H1N1)	i. Known human-to-human aerosol transmission	Upper respiratory: i. Viral RNA in pharyngeal and nasal swabs. [39]	i. Nosocomial transmission [39]
		Lower respiratory: i. Causes human respiratory disease [10]	
*Paramyxoviridae* i. Hendra virus ii. Nipah virus	i. ii. Suspected aerosol transmission i. NHPs infected by aerosol administration [40]	Upper respiratory: ii. Isolated from human nasal and throat secretions [41]	ii. Nosocomial transmission [42]
		Lower respiratory: i, ii Causes human respiratory disease.i. Viral antigen in human lung [43]	

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
