# Peer review of "Nosocomial Transmission of Emerging Viruses via Aerosol-Generating Medical Procedures"

_viruses, 2019, doi:10.3390/v11100940_

Round 1
Reviewer 1 Report
This review deals with a subject interesting and still topical: the risk of nosocomial transmission of viral infections through aerosol-generating medical procedures (AGMPs) in particular to healthcare workers (HCWs) attending infected patients. The main aim of this review is to investigate which AGMPs and which viruses are most at risk. As for viruses, the authors focus on emerging viruses highly pathogenic, for which there are limited prophylactic or therapeutic countermeasures. The results reported are based on some, limited, bibliographic citations. At the end the authors suggest some experimental approaches to better understand the risks associated with difference AGMPs and viruses.
As previously stated, the literature citations in support of the content of this review are scarce either in the section of the manuscript concerning the emerging viruses and their transmissibility by aerosol and in the last part in the section 4 concerning designing experiments and assessing risk.
In table 1, the first 5 AGMP are accompanied by the association with nosocomial virus transmission and related bibliographic entry unlike the following ones. Also in this regard, bibliographical references would be appropriate or the indication “not known as regards emerging viruses” or a similar sentence. The difference between AGMPs that mechanically create and disperse aerosols and procedures that induce the patient to produce aerosols is well explained in the text, the Figure is pretty but it is less clear.
Some bibliographic citations, like those suggested below, should be added (following the instructions of the journal).
Simonds AK, Hanak A, Chatwin M, Morrell MJ, Hall A, Parker KH, et al. Evaluation of droplet dispersion during non-invasive ventilation, oxygen therapy, nebulizer treatment and chest physiotherapy in clinical practice: implications for management of pandemic influenza and other airborne infections. Health Technol Assess 2010;14(46):131–172. [section 4]
Thompson K-A, Pappachan JV, Bennett AM, Mittal H, Macken S, Dove BK, et al. (2013) Influenza Aerosols in UK Hospitals during the H1N1 (2009) Pandemic – The Risk of Aerosol Generation during Medical Procedures. PLoS ONE 8(2): e56278. https://doi.org/10.1371/journal.pone.0056278 [section 4]
Robert A. Fowler, Cameron B. Guest, Stephen E. Lapinsky, William J. Sibbald, Marie Louie, Patrick Tang, Andrew E. Simor, and Thomas E. Stewart. Transmission of Severe Acute Respiratory Syndrome during Intubation and Mechanical Ventilation. Am J Respir Crit Care Med Vol 169. pp 1198–1202, 2004 https://doi.org/10.1164/rccm.200305-715OC [section 3.3]
Francoise M. Blacherea, , William G. Lindsleya, Cynthia M. McMillena, Donald H. Beezholda, Edward M. Fisherc, Ronald E. Shafferc, and John D. Notia. Assessment of influenza virus exposure and recovery from contaminated surgical masks and N95 respirators. J Virol Methods. 2018; 260: 98–106. doi:10.1016/j.jviromet.2018.05.009. [section 4]
Rule AM, Apau O, Ahrenholz SH, Brueck SE, Lindsley WG, de Perio MA, et al. (2018) Healthcare personnel exposure in an emergency department during influenza season. PLoS ONE 13 (8): e0203223. https://doi.org/10.1371/journal. pone.0203223 [section 4]
Jennifer C. Hunter, Duc Nguyen, Bashir Aden, Zyad Al Bandar, Wafa Al Dhaheri,
Kheir Abu Elkheir, Ahmed Khudair, Mariam Al Mulla, Feda El Saleh, Hala Imambaccus,
Nawal Al Kaabi, Farrukh Amin Sheikh, Jurgen Sasse, Andrew Turner, Laila Abdel Wareth,
Stefan Weber, Asma Al Ameri, Wesal Abu Amer, Negar N. Alami, Sudhir Bunga, Lia M. Haynes,
Aron J. Hall, Alexander J. Kallen, David Kuhar, Huong Pham, Kimberly Pringle, Suxiang Tong,
Brett L. Whitaker, Susan I. Gerber, Farida Ismail Al Hosani. Transmission of Middle East
Respiratory Syndrome Coronavirus Infections in Healthcare Settings, Abu Dhabi
Emerging Infectious Diseases • www.cdc.gov/eid • Vol. 22, No. 4, April 2016 647-656 [section 3.3]
Raymond Tellier. Review of aerosol transmission of influenza A virus. Emerging Infectious Diseases www.cdc.gov/eid Vol. 12, No. 11, November 2006 [section 3.5]
Bertran K, Balzli C, Kwon Y-K, Tumpey TM, Clark A, Swayne DE. Airborne transmission of highly pathogenic influenza virus during processing of infected poultry. Emerg Infect Dis. 2017 Nov [date cited]. https://doi.org/10.3201/eid2311.170672 [section 3.5]
Mathilde Richard, Ron A.M. Fouchier. Influenza A virus transmission via respiratory aerosols or droplets as it relates to pandemic potential. FEMS Microbiol Rev. 2016 Jan; 40(1): 68–85. Published online 2015 Sep 17. doi: 10.1093/femsre/fuv039 [section 3.5]
Huazhong Chen, Shelan Liu, Jun Liu, Chengliang Chai, Haiyan Mao, Zhao Yu, Yuming Tang, Geqin Zhu, Haixiao X. Chen, Chengchu Zhu, Hui Shao, Shuguang Tan, Qianli Wang, Yuhai Bi, Zhen Zou, Guang Liu, Tao Jin, Chengyu Jiang, George F. Gao, Malik Peiris, Hongjie Yu, Enfu Chen. Nosocomial Co-Transmission of Avian Influenza A(H7N9) and A(H1N1)pdm09 Viruses between 2 Patients with Hematologic Disorders. Emerg Infect Dis. 2016 Apr; 22(4): 598–607. doi: 10.3201/eid2204.151561 [section 3.5]
Chun-Fu Fang, Mai-Juan Ma, Bing-Dong Zhan, Shi-Ming Lai, Yi Hu, Xiao-Xian Yang, Jing Li, Guo-Ping Cao, Jing-Jing Zhou, Jian-Min Zhang, Shuang-Qing Wang, Xiao-Long Hu, Yin-Jun Li, Xiao-Xiao Wang, Wei Cheng, Hong-Wu Yao, Xin-Lou Li, Huai-Ming Yi, Wei-Dong Xu, Jia-Fu Jiang, Gregory C Gray, Li-Qun Fang, En-Fu Chen, Wu-Chun Cao. Nosocomial transmission of avian influenza A (H7N9) virus in China: epidemiological investigation. BMJ. 2015; 351: h5765. Published online 2015 Nov 19. doi: 10.1136/bmj.h5765 [section 3.5]
Janna Lietz, Claudia Westermann, Albert Nienhaus, Anja Schablon. The Occupational Risk of Influenza A (H1N1) Infection among Healthcare Personnel during the 2009 Pandemic: A Systematic Review and Meta-Analysis of Observational Studies. PLoS One. 2016; 11(8): e0162061. Published online 2016 Aug 31. doi: 10.1371/journal.pone.0162061 [section 3.5]
Constance Schultsz, Vo Cong Dong, Nguyen Van Vinh Chau, Nguyen Thi Hanh Le, Wilina Lim, Tran Tan Thanh, Christiane Dolecek, Menno D. de Jong, Tran Tinh Hien, Jeremy Farrar. Avian Influenza H5N1 and Healthcare Workers. Emerg Infect Dis. 2005 Jul; 11(7): 1158–1159. doi: 10.3201/eid1107.050070 [section 3.5]
Chun-Fu Fang, Mai-Juan Ma, Bing-Dong Zhan, Shi-Ming Lai, Yi Hu, Xiao-Xian Yang, Jing Li, Guo-Ping Cao, Jing-Jing Zhou, Jian-Min Zhang, Shuang-Qing Wang, Xiao-Long Hu, Yin-Jun Li, Xiao-Xiao Wang, Wei Cheng, Hong-Wu Yao, Xin-Lou Li, Huai-Ming Yi, Wei-Dong Xu, Jia-Fu Jiang, Gregory C Gray, Li-Qun Fang, En-Fu Chen, Wu-Chun Cao. Nosocomial transmission of avian influenza A (H7N9) virus in China: epidemiological investigation. BMJ. 2015; 351: h5765. Published online 2015 Nov 19. doi: 10.1136/bmj.h5765 [section 3.5]
Author Response
Dear Editors,
We are pleased with the thorough reviews by the reviewers and appreciate their recommended additions to our manuscript. We have added references and changes as recommended by the reviewers and have included our response to their comments below.
Sincerely yours, also on behalf of the co-authors,
Vincent Munster, PhD
Chief, Virus Ecology Unit
Laboratory of Virology, Rocky Mountain Laboratories, NIAID/NIH
903 South 4th street, Hamilton, MT 59840
406-375-7489
Reviewer 1 Comments:
This review deals with a subject interesting and still topical: the risk of nosocomial transmission of viral infections through aerosol-generating medical procedures (AGMPs) in particular to healthcare workers (HCWs) attending infected patients. The main aim of this review is to investigate which AGMPs and which viruses are most at risk. As for viruses, the authors focus on emerging viruses highly pathogenic, for which there are limited prophylactic or therapeutic countermeasures. The results reported are based on some, limited, bibliographic citations. At the end the authors suggest some experimental approaches to better understand the risks associated with difference AGMPs and viruses.
As previously stated, the literature citations in support of the content of this review are scarce either in the section of the manuscript concerning the emerging viruses and their transmissibility by aerosol and in the last part in the section 4 concerning designing experiments and assessing risk.
Response: We appreciate the reviewer’s comments and suggestion for additional citations, especially in section 4. We have updated that section as further discussed below in regards to reviewer 2’s comments with additional references as provided by reviewer 1.
In table 1, the first 5 AGMP are accompanied by the association with nosocomial virus transmission and related bibliographic entry unlike the following ones. Also in this regard, bibliographical references would be appropriate or the indication “not known as regards emerging viruses” or a similar sentence. The difference between AGMPs that mechanically create and disperse aerosols and procedures that induce the patient to produce aerosols is well explained in the text, the Figure is pretty but it is less clear.
Response: We appreciate the reviewer’s comments about Table 1 and Figure 1. We have edited the table as also recommended by reviewer 2 to make the table more balanced. We have edited the label of Figure 1 to include the following additional details for clarification:
Figure 1. Potential types of aerosol-generating medical procedures (AGMPs)
AGMPs can be divided into procedures that induce the patient to produce / aerosols and procedures that mechanically generate aerosols themselves
Some bibliographic citations, like those suggested below, should be added (following the instructions of the journal).
Simonds AK, Hanak A, Chatwin M, Morrell MJ, Hall A, Parker KH, et al. Evaluation of droplet dispersion during non-invasive ventilation, oxygen therapy, nebulizer treatment and chest physiotherapy in clinical practice: implications for management of pandemic influenza and other airborne infections. Health Technol Assess 2010;14(46):131–172. [section 4]
Thompson K-A, Pappachan JV, Bennett AM, Mittal H, Macken S, Dove BK, et al. (2013) Influenza Aerosols in UK Hospitals during the H1N1 (2009) Pandemic – The Risk of Aerosol Generation during Medical Procedures. PLoS ONE 8(2): e56278. https://doi.org/10.1371/journal.pone.0056278 [section 4]
Robert A. Fowler, Cameron B. Guest, Stephen E. Lapinsky, William J. Sibbald, Marie Louie, Patrick Tang, Andrew E. Simor, and Thomas E. Stewart. Transmission of Severe Acute Respiratory Syndrome during Intubation and Mechanical Ventilation. Am J Respir Crit Care Med Vol 169. pp 1198–1202, 2004 https://doi.org/10.1164/rccm.200305-715OC [section 3.3]
Francoise M. Blacherea, , William G. Lindsleya, Cynthia M. McMillena, Donald H. Beezholda, Edward M. Fisherc, Ronald E. Shafferc, and John D. Notia. Assessment of influenza virus exposure and recovery from contaminated surgical masks and N95 respirators. J Virol Methods. 2018; 260: 98–106. doi:10.1016/j.jviromet.2018.05.009. [section 4]
Rule AM, Apau O, Ahrenholz SH, Brueck SE, Lindsley WG, de Perio MA, et al. (2018) Healthcare personnel exposure in an emergency department during influenza season. PLoS ONE 13 (8): e0203223. https://doi.org/10.1371/journal. pone.0203223 [section 4]
Jennifer C. Hunter, Duc Nguyen, Bashir Aden, Zyad Al Bandar, Wafa Al Dhaheri,
Kheir Abu Elkheir, Ahmed Khudair, Mariam Al Mulla, Feda El Saleh, Hala Imambaccus,
Nawal Al Kaabi, Farrukh Amin Sheikh, Jurgen Sasse, Andrew Turner, Laila Abdel Wareth,
Stefan Weber, Asma Al Ameri, Wesal Abu Amer, Negar N. Alami, Sudhir Bunga, Lia M. Haynes,
Aron J. Hall, Alexander J. Kallen, David Kuhar, Huong Pham, Kimberly Pringle, Suxiang Tong,
Brett L. Whitaker, Susan I. Gerber, Farida Ismail Al Hosani. Transmission of Middle East
Respiratory Syndrome Coronavirus Infections in Healthcare Settings, Abu Dhabi
Emerging Infectious Diseases • www.cdc.gov/eid • Vol. 22, No. 4, April 2016 647-656 [section 3.3]
Raymond Tellier. Review of aerosol transmission of influenza A virus. Emerging Infectious Diseases www.cdc.gov/eid Vol. 12, No. 11, November 2006 [section 3.5]
Bertran K, Balzli C, Kwon Y-K, Tumpey TM, Clark A, Swayne DE. Airborne transmission of highly pathogenic influenza virus during processing of infected poultry. Emerg Infect Dis. 2017 Nov [date cited]. https://doi.org/10.3201/eid2311.170672 [section 3.5]
Mathilde Richard, Ron A.M. Fouchier. Influenza A virus transmission via respiratory aerosols or droplets as it relates to pandemic potential. FEMS Microbiol Rev. 2016 Jan; 40(1): 68–85. Published online 2015 Sep 17. doi: 10.1093/femsre/fuv039 [section 3.5]
Huazhong Chen, Shelan Liu, Jun Liu, Chengliang Chai, Haiyan Mao, Zhao Yu, Yuming Tang, Geqin Zhu, Haixiao X. Chen, Chengchu Zhu, Hui Shao, Shuguang Tan, Qianli Wang, Yuhai Bi, Zhen Zou, Guang Liu, Tao Jin, Chengyu Jiang, George F. Gao, Malik Peiris, Hongjie Yu, Enfu Chen. Nosocomial Co-Transmission of Avian Influenza A(H7N9) and A(H1N1)pdm09 Viruses between 2 Patients with Hematologic Disorders. Emerg Infect Dis. 2016 Apr; 22(4): 598–607. doi: 10.3201/eid2204.151561 [section 3.5]
Chun-Fu Fang, Mai-Juan Ma, Bing-Dong Zhan, Shi-Ming Lai, Yi Hu, Xiao-Xian Yang, Jing Li, Guo-Ping Cao, Jing-Jing Zhou, Jian-Min Zhang, Shuang-Qing Wang, Xiao-Long Hu, Yin-Jun Li, Xiao-Xiao Wang, Wei Cheng, Hong-Wu Yao, Xin-Lou Li, Huai-Ming Yi, Wei-Dong Xu, Jia-Fu Jiang, Gregory C Gray, Li-Qun Fang, En-Fu Chen, Wu-Chun Cao. Nosocomial transmission of avian influenza A (H7N9) virus in China: epidemiological investigation. BMJ. 2015; 351: h5765. Published online 2015 Nov 19. doi: 10.1136/bmj.h5765 [section 3.5]
Janna Lietz, Claudia Westermann, Albert Nienhaus, Anja Schablon. The Occupational Risk of Influenza A (H1N1) Infection among Healthcare Personnel during the 2009 Pandemic: A Systematic Review and Meta-Analysis of Observational Studies. PLoS One. 2016; 11(8): e0162061. Published online 2016 Aug 31. doi: 10.1371/journal.pone.0162061 [section 3.5]
Constance Schultsz, Vo Cong Dong, Nguyen Van Vinh Chau, Nguyen Thi Hanh Le, Wilina Lim, Tran Tan Thanh, Christiane Dolecek, Menno D. de Jong, Tran Tinh Hien, Jeremy Farrar. Avian Influenza H5N1 and Healthcare Workers. Emerg Infect Dis. 2005 Jul; 11(7): 1158–1159. doi: 10.3201/eid1107.050070 [section 3.5]
Chun-Fu Fang, Mai-Juan Ma, Bing-Dong Zhan, Shi-Ming Lai, Yi Hu, Xiao-Xian Yang, Jing Li, Guo-Ping Cao, Jing-Jing Zhou, Jian-Min Zhang, Shuang-Qing Wang, Xiao-Long Hu, Yin-Jun Li, Xiao-Xiao Wang, Wei Cheng, Hong-Wu Yao, Xin-Lou Li, Huai-Ming Yi, Wei-Dong Xu, Jia-Fu Jiang, Gregory C Gray, Li-Qun Fang, En-Fu Chen, Wu-Chun Cao. Nosocomial transmission of avian influenza A (H7N9) virus in China: epidemiological investigation. BMJ. 2015; 351: h5765. Published online 2015 Nov 19. doi: 10.1136/bmj.h5765 [section 3.5]
Response: We greatly appreciate the reviewer’s thoughtful suggestions for additional references and have included all of the above references except for Bertran et al. 2017 because it was outside of the healthcare setting.
Reviewer 2 Report
In this review by Judson and Munster, the authors describe the scope of known aerosol-generating medical procedures that might lead to nosocomial transmission of a group of pathogens they term high-risk viruses. The review is generally well-written and the topic is of public health relevance and applicable for the journal. Table 2 and the associated text that supports this table represent the most comprehensive and most well-referenced section of the article. However, there are other areas of this review that would benefit from additional reference and context, as described below.
Comments:
Section 4 (“designing experiments and assessing risk”) requires additional references to support the text; there’s currently only one reference for the entire section. Critically, the authors should provide additional supportive references to support the multiple air sampling techniques used for virus collection (lines 266-7), and additional supportive references to direct readers for more information regarding the particle sizers mentioned (lines 267-70). However, most sentences in this section in its entirety would benefit from supportive references to provide the reader with additional information or context. The focus of this review is on high-risk viruses, that is, viruses (many from zoonotic reservoirs) with limited prophylactic or therapeutic countermeasures. It is possible that many health care workers potentially exposed to virus-laden aerosols would be working in sub-optimal clinical environments (i.e. facilities with unreliable power/water access, reduced or absence HVAC infrastructure, with differential access to appropriate personal protective equipment, etc). It would be of great benefit to the reader if the authors could describe and cite in more depth the specific environment they mean by “nosocomial” when describing aerosol-generating procedures that might pose a health threat to health care workers. Table 1 and section 2 seem to pertain more to advanced healthcare hospital settings. Are there differences in equipment/procedures/practices by health care workers when caring for patients in advanced hospital settings vs field clinics that would raise/lower the risk of aerosol-generating procedures? Similar to comment 2, it would be of benefit for the authors to mention the role (if any) environmental conditions such as temperature and humidity play in aerosol-generating medical procedures, and if the specific nosocomial environment where these procedures are taking place have adequate regulation of these factors. Table 1, the table would likely be more balanced by removing the third column (association with nosocomial virus transmission), indicating this information with a footnote, and having the table stand with just the first two columns that are more comprehensive. Section 3. The first paragraph (lines 120-135) would benefit greatly from additional references supporting the viruses mentioned, even in the context of their exclusion from the definition of high-risk pathogens. A review article such as PMID 21497126, Pedrosa and Cardoso, or something along those lines would be helpful, should the reader be interested in learning more about the pathogens mentioned in this text.Author Response
Reviewer 2 Comments:
In this review by Judson and Munster, the authors describe the scope of known aerosol-generating medical procedures that might lead to nosocomial transmission of a group of pathogens they term high-risk viruses. The review is generally well-written and the topic is of public health relevance and applicable for the journal. Table 2 and the associated text that supports this table represent the most comprehensive and most well-referenced section of the article. However, there are other areas of this review that would benefit from additional reference and context, as described below.
Comments:
Section 4 (“designing experiments and assessing risk”) requires additional references to support the text; there’s currently only one reference for the entire section. Critically, the authors should provide additional supportive references to support the multiple air sampling techniques used for virus collection (lines 266-7), and additional supportive references to direct readers for more information regarding the particle sizers mentioned (lines 267-70). However, most sentences in this section in its entirety would benefit from supportive references to provide the reader with additional information or context.
Response: We appreciate the reviewer’s comments and have updated Section 4 to include more details and references regarding air sampling techniques, particle sizers, and experimental design, as follows:
“Air sampling for viruses during AGMPs performed on patients would provide the most clinically relevant data. Multiple air sampling techniques for viruses now exist. Researchers can use both solid and liquid impactors to sample and recover aerosolized viruses [42]. Personally worn bioaerosal samplers and stationary room samplers were used to detect influenza A virus RNA in an emergency department [43]. Researchers also used aerosol samplers to determine the amount of influenza A (H1N1) RNA in aerosols in the vicinity of patients while AGMPs were being performed [44]. This allowed the authors to determine which procedures were associated with a higher concentration of viral RNA [44]. In order to determine whether viable virus is present in aerosols, virus isolation could be performed from air samples and further quantification could occur through titrations or plaque assays [42]. One group used a simulated aerosol chamber to demonstrate that viable influenza virus A could be extracted from surgical masks and N95 respirators [45]. Particle sizers may also be used in experimental settings to characterize the size and dispersal of aerosols. One group used particle sizers to measure the size and travel distance of aerosols from patients who underwent AGMPs such as nebulizer treatment and NIV [46]. Determining the quantity of viable virus expelled from certain patients during AGMPs could help determine phenomena like super-spreading events, while understanding aerosol characteristics such as particle size could elucidate mechanisms of transmission.”
The focus of this review is on high-risk viruses, that is, viruses (many from zoonotic reservoirs) with limited prophylactic or therapeutic countermeasures. It is possible that many health care workers potentially exposed to virus-laden aerosols would be working in sub-optimal clinical environments (i.e. facilities with unreliable power/water access, reduced or absence HVAC infrastructure, with differential access to appropriate personal protective equipment, etc). It would be of great benefit to the reader if the authors could describe and cite in more depth the specific environment they mean by “nosocomial” when describing aerosol-generating procedures that might pose a health threat to health care workers. Table 1 and section 2 seem to pertain more to advanced healthcare hospital settings. Are there differences in equipment/procedures/practices by health care workers when caring for patients in advanced hospital settings vs field clinics that would raise/lower the risk of aerosol-generating procedures? Similar to comment 2, it would be of benefit for the authors to mention the role (if any) environmental conditions such as temperature and humidity play in aerosol-generating medical procedures, and if the specific nosocomial environment where these procedures are taking place have adequate regulation of these factors.
Response: We appreciate the reviewer bringing attention to the important point of how environmental conditions and healthcare settings influence risk. We have included the following 2 paragraphs in Section 4 to highlight this issue:
“Multiple environmental factors influence the viability of aerosolized viruses, including relative humidity, temperature, UV radiation, and gas composition of the air [42]. These factors affect viruses differently, and so it is important to consider the environments where emerging viruses exist and where AGMPs are being performed. The aforementioned AGMPs include those that are both ubiquitous, such as CPR or manual ventilation, as well as those that are limited to advanced healthcare settings, such as bronhoscopy. The risk of transmission from AGMPs may be very different in a field-based treatment unit than from a tertiary care hospital. Additionally, some healthcare settings, such as hospitals with biocontainment units, include additional air handling systems that can limit aerosol exposure to HCWs [49]. Therefore, the risk of nosocomial transmission via AGMPs varies greatly based on the environment, and this must be considered when designing experiments.”
Table 1, the table would likely be more balanced by removing the third column (association with nosocomial virus transmission), indicating this information with a footnote, and having the table stand with just the first two
columns that are more comprehensive.
Response: We have edited the table as recommended by the reviewer
Section 3. The first paragraph (lines 120-135) would benefit greatly from additional references supporting the viruses mentioned, even in the context of their exclusion from the definition of high-risk pathogens. A review article such as PMID 21497126, Pedrosa and Cardoso, or something along those lines would be helpful, should the reader be interested in learning more about the pathogens mentioned in this text.
Response: We appreciate the reviewer’s suggestion and have added the following reference to a review article about the mentioned viruses and airborne precautions.
Shiu EY, Leung NH, Cowling BJ. Controversy around airborne versus droplet transmission of respiratory viruses. Current Opinion in Infectious Diseases. 2019;32(4):372–379. doi: 10.1097/QCO.0000000000000563.